# Towards New Scaffolds for Antimicrobial Activity—In Silico/In Vitro Workflow Introducing New Lead Compounds

**DOI:** 10.3390/antibiotics14010011

**Published:** 2024-12-27

**Authors:** Maria Mangana, George Lambrinidis, Ioannis K. Kostakis, Ioanna Kalpaktsi, Marina Sagnou, Chrysoula Nicolaou, Emmanuel Mikros, Stylianos Chatzipanagiotou, Anastasios Ioannidis

**Affiliations:** 1Department of Clinical Microbiology, Athens Medical School, Aeginition Hospital, 11528 Athens, Greece; mariamgn91@gmail.com (M.M.); chrysoula.nikolaou@gmail.com (C.N.); schatzipa@gmail.com (S.C.); 2Division of Pharmaceutical Chemistry, Department of Pharmacy, School of Health Sciences, National and Kapodistrian University of Athens, 15771 Athens, Greece; ikkostakis@pharm.uoa.gr (I.K.K.); jkalpaktsi@gmail.com (I.K.); mikros@pharm.uoa.gr (E.M.); 3Institute of Biosciences & Application, NCSR “Demokritos”, 15310 Athens, Greece; sagnou@bio.demokritos.gr; 4Department of Science and Mathematics, Deree-The American College of Greece, 15342 Athens, Greece; 5Department of Nursing, Faculty of Health Sciences, University of Peloponnese, 22100 Tripolis, Greece

**Keywords:** ligand-based drug design, multidrug-resistant bacteria, antimicrobial activity, chemical libraries, lead optimization

## Abstract

**Background/Objectives**: The rapid evolution of bacterial resistance and the high cost of drug development have attributed greatly to the dearth in drug design. Computational approaches and natural product exploitation offer potential solutions to accelerate drug discovery. **Methods**: In this research article, we aimed to identify novel antibacterial hits. For the in silico studies, molecular scaffolds from the in-house chemical library of the Department of Pharmacy of Athens (Pharmalab) and the National Cancer Institute (NCI) were screened and selected for further experimental procedures. Compounds from both libraries that were not previously screened for their antimicrobial properties were tested in vitro against Gram-positive and Gram-negative bacterial strains. The microdilution method was used to determine the minimum inhibitory concentrations (MICs). **Results**: In silico screening identified twenty promising molecules from the NCI and seven from the Pharmalab databases. The unexplored compounds for their antibacterial activity can be characterized as weak strain-specific antimicrobials. The **NSC 610491** and **NSC 610493** were active against *Staphylococcus aureus* (MIC: 25 and 12.5 µg/mL, respectively) and methicillin-resistant *S. aureus* (MRSA) (MIC: 50 and 12.5 µg/mL, respectively). Six out of seven hydroxytyrosol (HTy) compounds were moderately active (MIC: 25–50 µg/mL) against *S. aureus*, MRSA and *Enterococcus faecalis*. For the Gram-negative bacteria, no activity was detected (≥100 µg/mL). **Conclusions**: The tested scaffolds could be considered as promising candidates for novel antimicrobials with improvements. Further experimentation is required to assess mechanisms of action and evaluate the efficacy and safety.

## 1. Introduction

The current trend in human mortality highly depends on drug-resistant microbial infections. The efficacy of antibiotics has begun to fade in the rise of an elite class of “superbugs”. In 2021, estimated 1.14 million deaths worldwide were attributed to bacterial antimicrobial resistance (AMR) [1], underscoring the escalating global health crisis posed by drug-resistant bacteria despite advancements in antimicrobial stewardship and infection control measures. In the US alone, more than 2.8 million people are infected annually by multidrug-resistant (MDR) pathogens, leading to more than 35,000 deaths, while in Europe, the toll exceeds 33,000 cases [2]. Reports on healthcare-associated infections in the U.S. healthcare facilities indicate that *Escherichia coli* is the most frequently identified pathogen, followed by *Staphylococcus aureus*, *Klebsiella* spp., and *Pseudomonas aeruginosa*, while *Enterococcus faecalis* is the most prevalent in long-term acute care hospitals [3]. Finally, the increasing incidences of multidrug-resistant *Acinetobacter baumannii* strains in surgical trauma infections, ventilator-associated pneumonia, and bacteremia in intensive care units (ICUs) have raised concerns about the effectiveness in the management of this pathogen [4]. Moreover, the emergence of the first plasmid-mediated gene conferring resistance to polymyxin (mcr-1) has further complicated the resistance crisis [5]. This development renders colistin (polymyxin E), the last viable therapeutic option for carbapenem-resistant Enterobacteriaceae (CRE), ineffective.

Drug design has not kept pace with this evolving threat, leading to a critical shortage of effective treatments for serious infections. Over the last decade, combinations of chemical compounds with known antibiotics have been extensively investigated as therapeutic regimens to treat diseases caused by MDR bacteria. The dearth of innovative antibiotics, featuring novel chemical classes or mechanisms of action, has been a hallmark of the antibiotic discovery landscape in recent years [6]. Developing new antibiotics is a complex, time-consuming and costly endeavor that faces several multifactorial hurdles [7,8]. In general, drug development demands extensive research, clinical trials, and regulatory approval. Patent protection provides temporary market exclusivity, while navigating the complex regulatory landscape may further increase the overall cost of drug development. Moreover, drugs targeting resistant pathogens may have limited market potential because the rapid evolution of microorganisms enables them to quickly develop resistance, thus reducing the drugs’ long-term effectiveness in disease resolution [9]. Very few new classes of antibiotics have been discovered in the last 40 years, with resistance often emerging within 5 years of a drug’s market entry; hence, the efforts of big pharmaceutical companies to this end have been diminished [10]. Horizontal gene transfer accelerates the spread of resistance in pathogens, while the misuse of currently applied antimicrobials significantly contributes to the development and spread of resistance. These challenges collectively contribute to the high cost of drug development and can discourage pharmaceutical companies from investing in certain therapeutic areas, particularly those with high risk of resistance [9,11].

The era of pandrug-resistant pathogens necessitates the rapid discovery and development of novel antimicrobials with improved efficacy and reduced toxicity. To address these challenges, computational approaches have emerged as powerful tools for accelerating the drug discovery process [12]. In silico methods have gained significant traction in recent years [13]. By leveraging computational power, virtual screening (VS) enables researchers to rapidly assess the potential of large chemical libraries against biological targets [14,15]. VS methods can accurately predict protein−ligand binding poses, often outperforming traditional screening approaches. Additionally, VS has been proven effective in identifying novel chemical scaffolds suitable for drug development. Mining into the resulting target-specific information and chemotypes in thorough post-screening analysis, researchers anticipate the ability to increase the likelihood of discovering drug candidates at an early stage [16]. These lead compounds are then optimized through iterative circles of synthesis and biological evaluation.

Pharmaceutical companies recommend that future efforts towards identifying novel antimicrobials should focus on testing libraries that sample a broader diversity of chemical space, incorporating compounds with different physicochemical properties—possibly through diversity-oriented composition libraries—and natural products that have not yet been extensively screened for antimicrobial activity [10]. Natural resources, such as plants, have emerged as a promising source of novel antimicrobial molecules, as they possess a rich reservoir of bioactive compounds capable of targeting multiple mechanisms of antimicrobial resistance [17]. Natural product libraries typically consist of a large number of compounds, including those classified as pan-assay interfering and nonspecific compounds (PAINSs), which often exhibit pleiotropic activity (e.g., antimicrobial, anti-inflammatory, and anticancer), making them a valuable tool for a wide range of tests [18]. An interesting example of PAINS is the natural product curcumin, the main polyphenolic curcuminoid isolated from turmeric (*Curcuma longa*). Curcumin has been extensively studied for its pleiotropic properties [19]. In an effort to expand the chemical space for our scaffolds search, we attempted to screen for compounds with structural similarity to curcumin and demethoxycurcumin which possess known antibacterial activity [20,21,22,23].

The present study aims to discover new candidate antimicrobial compounds through the application of computational drug design methods. Specifically, in silico screening techniques were utilized in large compound databases such as the National Cancer Institute (NCI) and the Pharmalab in-house chemical libraries. Using ligand-based design approaches, candidate compounds with potential antimicrobial activity were selected. These compounds were then subjected to biological evaluation to confirm their antimicrobial properties. The ultimate goal of the research is the discovery of new hits with improved antimicrobial potency and selectivity. By combining computational methods with experimental validation, we seek to accelerate the hit-to-lead discovery process.

## 2. Results

### 2.1. Similarity Search

Initially, we searched the CHEMBL database to retrieve compounds tested against *E. coli* [24]. From a total of nearly 4000 entries, the first 8 CHEMBL addressed molecules with the best minimum inhibitory concentration (MIC) were selected as query molecules for similarity search (Figure 1). Additionally, we included the parent molecule of curcumin as well as demethoxycurcumin, which is structurally related to curcumin and naturally occurring in the curcuminoid natural and commercial extract [19] (Figure 1). This addition is part of an ongoing project on recently synthesized curcumin analogues exhibiting antimicrobial activity (unpublished data). Thus, a total number of ten reference molecules is investigated herein.

The ROCS (Rapid Overlay of Chemical Structures) software of the OpenEye Scientific Software^®^ (ROCS 3.5.1.1: OpenEye Scientific Software, Santa Fe, NM, USA; http://www.eyesopen.com) was utilized [25]. In our study, the degree of similarity between the query molecules (NCI/Pharmalab) and the reference molecules (CHEMBL/CURCUMINS) was evaluated by means of the Tanimoto Combo score which was set to be greater than 1.0. For each of the ten selected query compounds, there were twenty different NCI compounds with matching similarity. Afterwards, each of these twenty molecules was individually screened in the CHEMBL online database to determine whether it has been previously tested for antimicrobial activity. This was achieved by proceeding to an initial similarity check at 95%, and in the absence of results, the remaining percentages of similarity were also considered (Table 1).

From Table 1, it was concluded that only seven compounds from the NCI chemical library, namely the **NSC 2807**, **NSC 34616**, **NSC 63326**, **NSC 118990**, **NSC 400770**, **NSC 610491**, and **NSC 610493** (Figure 2), showed potential for further investigation of their antimicrobial activity.

All NCI compounds were also searched in the PubChem platform (https://pubchem.ncbi.nlm.nih.gov/ (accessed on 10 September 2022)) to obtain any additional information on their structure and activity. **NSC 610491**, a ciprofloxacin analog, has been tested for anticancer properties in vivo. Another ciprofloxacin-related compound, **NSC 610492**, is patented for its antimicrobial activity (EP-0155244-A3) and was, therefore, excluded from further testing. **NSC 610492** (1-ethyl-6-fluoro-7-(1-pyrrole-1-yl)-1,4-dihydro-4-oxoquinoline-3-carboxylic acid) is a quinolone carboxylic acid derivative of the ’oxacin’ family, showing potent activity against Gram-positive bacteria, Enterobacteriaceae, *Proteus* and *Pseudomonas* spp. Its antimicrobial efficacy is superior to that of nalidixic acid, piroimidic acid and pipemic acid, and its antimicrobial spectrum is also slightly wider than the one of enoxacin. It is a potent fluorinated antibacterial quinolone widely used in the management of urinary tract infections and systemic infections. **NSC 610493**, another ciprofloxacin related structure, has been tested for anticancer and antiretroviral properties, while **NSC 63326** has been tested for antiretroviral activity. However, due to the current NCI policy limiting the distribution of samples from the Open Chemicals Repository Collection, which includes all vialed samples and plated sets, to investigators outside the United States and U.S. Territories, further investigation of additional compounds from NCI was restricted. As a result, our efforts were focused on screening compounds from Pharmalab, our in-house chemical library of natural and synthetic molecules. Using the same dataset of ten query molecules, Tanimoto Combo scores below 1.5 were identified for all compounds. To expand our selection range and potentially identify new lead compounds, molecules with Tanimoto Combo scores close to 1.0 were also included in our search. Seven synthetic analogues of HTy were selected and further prepared for in vitro evaluation. Their Tanimoto scores are illustrated on Table 2.

The in silico screening workflow and the filters described previously were followed for the selection of the final virtual hits for the Pharmalab chemical library as well [26]. The screening process ultimately identified seven molecules from the NCI database (Figure 2) and seven molecules from the Pharmalab in-house chemical library (Figure 3).

### 2.2. In Vitro Antibacterial Activity

The in vitro antimicrobial activity of the tested compounds is expressed in MIC scores and is presented on Table 3. The compounds **NSC 610491** (25 µg/mL), **NSC 610493** (12.5 µg/mL) and the **NSC 63326**, **HTy1**, **HTy3**, **HTy4**, **HTy5**, and **HTy6** (50 µg/mL) exhibited moderate to good activity. Based on their activity against *S. aureus*, when they were tested against MRSA, only the **NSC 610491** (50 µg/mL) and the **NSC 610493** (12.5 µg/mL) exhibited moderate to good activity. Most compounds had limited to no activity against the *E. faecalis* VRE strain, with the exception of **HTy2** (25 µg/mL) and **HTy1**, **-3** and **-6** (50 µg/mL). With regards to the Gram-negative bacteria, only compound **HTy5** exhibited a noticeable activity against *P. aeruginosa* (50 µg/mL), whereas the rest of the compounds were inactive (≥100 µg/mL). In general, almost all of the HTy compounds showed activity against at least two different pathogens, indicating a rather broad-spectrum activity potential. It was only the compound **HTy7** with the least activity hits. Similarly, **NSC 610493** was the compound with the broadest spectrum of effect since it was active against five strains.

## 3. Discussion

The in vitro bacterial susceptibility profile of our lead compounds was collectively characterized by low to moderate antimicrobial activity, depending on the strain. Overall, HTy compounds showed higher efficacy than the NCI compounds, particularly against *S. aureus* and *E. faecalis*, whereas both groups of compounds exhibited limited activity against MRSA and *P. aeruginosa*.

More specifically, the **NSC 63326**, **HTy3**, and **HTy5** demonstrated moderate anti-Gram-positive activity, a trend which could be associated with their structural similarity to demethoxycurcumin. Demethoxycurcumin has been shown to be efficacious against *S. aureus*, MRSA, and *E. faecalis*, which is in accordance to our findings and suggests that similarity to known active agents may have a predictive value for antimicrobial potency [23,27,28]. The ciprofloxacin-related NCI compounds, **NSC 610491** and **NSC 610493**, also exhibited moderate to good activity against *S. aureus* and MRSA, reflecting the broad-spectrum activity profile of ciprofloxacin while potentially mitigating some resistance-related issues [29].

In addition to their antibacterial properties, several HTy analogues have previously demonstrated notable antifungal activity, as has been previously reported by our group [30]. Hydroxytyrosol (HT), the parent compound of HTy derivatives of our library, is a primary phenolic compound in *Olea europaea* extracts with established antimicrobial, antiviral, and antifungal effects. Our earlier work showed that HT analogues exhibit strong antifungal activity, likely due to fungal membrane permeabilization, with good biosafety against mammalian cells. This amphiphilic nature may explain their efficacy across both bacterial and fungal membranes, supporting their potential for pleiotropic activity [31]. The hydroxytyrosol fragments seem important for antimicrobial activity against *S. aureus* and *E. faecalis*, while on *P. aeruginosa* the presence of adamantane (**HTy1, -3, -4,** and **-6**) or the absence of the ester group diminishes activity. It is worth noticing that in good agreement with this notion, **HTy2** and **HTy5** are the most active of the hydroxytyrosol derivatives which may be related to their amphiphilic nature possessing a distinct polar head and a non-polar hydrocarbon chain.

Overall, our findings suggest that compounds with structural similarity to known antimicrobial agents, such as demethoxycurcumin and ciprofloxacin, exhibit enhanced efficacy. However, the moderate levels of antimicrobial activity observed highlight the need for further optimization. To address these limitations and advance the development of potent antimicrobial agents, our future research will focus on expanding the chemical space by screening a wider range of compounds, including natural products and synthetic molecules, with higher Tanimoto Combo scores to identify novel scaffolds with improved activity. Additionally, mechanistic studies need to be applied to delineate the precise mechanisms of action of these compounds to gain insights into their mode of antimicrobial activity and potential resistance mechanisms. Pharmacokinetic and pharmacodynamic studies will be considered to evaluate the pharmacokinetic properties, including absorption, distribution, metabolism, and excretion, as well as the pharmacodynamic effects of these compounds in relevant animal models. Finally, another aspect for future research by our team is the assessment of the safety profile of these compounds in vitro and in vivo to identify potential toxicity concerns and optimize their therapeutic window.

## 4. Materials and Methods

### 4.1. In Silico Studies

#### 4.1.1. Ligand Preparation

Two chemical libraries were evaluated in our project: (1) our in-house library of 2000 compounds (synthetic and natural products) named “Pharmalab” [32]; and (2) National Cancer Institute (NCI) Open Database Compounds containing ∼260,000 molecules (https://cactus.nci.nih.gov/download/nci/ (accessed on 10 September 2022)).

All ligands were prepared using the LigPrep module as implemented in Schrodinger Suite 2021. The final output file was passed through Omega Software (OMEGA 4.2.1.1: OpenEye Scientific Software, Santa Fe, NM, USA; http://www.eyesopen.com) to generate up to 200 conformers per ligand for further virtual similarity screening.

#### 4.1.2. Similarity Search

For similarity search, the ROCS software (ROCS 3.5.1.1: OpenEye Scientific Software, Santa Fe, NM; http://www.eyesopen.com) was utilized using the default parameters. For each query molecule, the whole database was passed through a similarity search.

### 4.2. In Vitro Evaluation

#### 4.2.1. Bacterial Strains and Culture Conditions

*S. aureus* (ATCC^®^ 25923^TM^, ATCC: American Type Culture Collection, 10801 University Blvd, Manassas, VA, USA), *P. aeruginosa* (ATCC^®^ 27853^TM^), and *E. coli* (ATCC^®^ 25922^TM^) were used as wild-type strains. The following clinical isolates from patients with urinary tract infection were also used: methicillin-resistant *S. aureus* (MRSA 2679 ST80), *A. baumannii*, and *K. pneumoniae*. The clinical strains were classified as susceptible, intermediate, or resistant by antibiotic susceptibility testing and following the EUCAST breakpoints (http://www.eucast.org/clinical_breakpoints) (AST, MicroScan^®^, Beckman Coulter, Inc., Brea, CA, USA; Appendix A). *A. baumannii* was resistant to all tested antibiotics but gentamicin, minocycline, tobramycin, tigecycline, and colistin, whereas *K. pneumoniae* was resistant to ampicillin, cefepime, and piperacillin.

Bacteria were stored as frozen stocks in 20% (*v*/*v*) glycerol at −20 °C and were routinely streaked on appropriate agar plates for single colony isolation. Prior to use, isolates were grown in Mueller-Hinton Broth (MHB) (Scharlau^TM^, Barcelona, Spain) under continuous agitation (200 rpm; 37 °C) overnight. Bacterial growth was evaluated by optical density (OD_600_) using a BioPhotometer plus 6132 (Eppendorf AG, Hamburg, Germany). All bacterial cultures were grown to late log phase (0.5–0.7 OD_600_) and re-suspended in MHB. The number of viable bacteria cells was calculated as colony-forming units (CFUs) per mL. Briefly, cells were harvested by centrifugation at 5000× *g* for 10 min, washed thrice with PBS and serially diluted. A 10 µL aliquot of each dilution was spotted onto MH agar (MHA) plates to determine CFU/mL. Only dilutions that yielded up to 100 CFUs were counted.

#### 4.2.2. Antimicrobial Susceptibility Testing

Growth inhibition was determined in flat-bottom 96-well plates after the preparation of a twofold serial dilution of the test analogs. The compounds ordered and shipped from the NCI were in the form of solid powders stored in aliquots. The synthesized compounds were isolated as solid powders and fully characterized by spectroscopic techniques as described elsewhere [31]. All analogs (NSC and HTy) were prepared in 5% dimethyl sulfoxide (DMSO) prior to use and stored at −20 °C. Fresh bacterial cultures were prepared in MHB as described above, and bacterial numbers were adjusted to 10^5^ CFU/mL. The MIC was determined following the EUCAST recommendations. Growth was evaluated with a spectrophotometer (DSX Dynex, Dynex Technologies Inc., Chantilly, VA, USA) by absorption at 600 nm after overnight incubation (18 h) at 37 °C. The MIC values were defined as the compound concentration that completely inhibited cell growth after incubation.

### 4.3. Statistical Analysis

All tests were performed in triplicates and replicated in two independent experiments; means from two independent experiments and standard deviations (SD) were calculated, unless otherwise stated.

## 5. Conclusions

In conclusion, HTy compounds, particularly **HTy3** and **HTy5**, demonstrated promising activity against *S. aureus* and *E. faecalis*, likely due to their structural similarity to demethoxycurcumin. The NCI compounds, especially **NSC 610491** and **NSC 610493**, showed moderate to good activity against *S. aureus* and MRSA, potentially leveraging the broad-spectrum activity of ciprofloxacin. The amphiphilic nature of the HTy compounds, reminiscent of hydroxytyrosol, may contribute to their efficacy against both bacterial and fungal membranes, suggesting potential for pleiotropic activity. Overall, our findings suggest that the tested scaffolds are promising candidates for further optimization and could lead to the development of novel antimicrobials. Further investigation into the structure-activity relationships of these compounds may lead to the development of novel antimicrobial agents with improved potency and selectivity. The moderate activity observed in many compounds highlights a need for structural modifications to enhance efficacy and broaden the spectrum of activity. Our aim for future research is to overcome the limitation of moderate structural similarity of the query molecules and the molecules in question, as interpreted by Tanimoto Combo scores close to 1.0. Thus, we aim to focus our efforts on screening natural and synthetic molecules from Pharmalab and the NCI chemical library targeting query molecules with Tanimoto Combo scores equal or greater than 1.5. Future research will focus on the following aspects: (i) assessing specific mechanisms of action; (ii) evaluating persistence; and (iii) determining the efficacy and safety of these compounds in vitro (e.g., human skin fibroblasts) and in vivo (animal models).

## Figures and Tables

**Figure 1 antibiotics-14-00011-f001:**
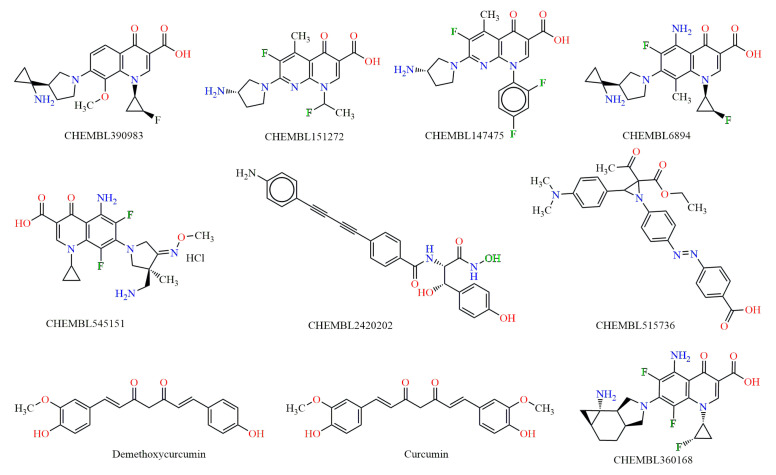
Chemical structures of the reference molecules (CHEMBL/CURCUMINS) for the similarity search. Curcumin (1E,6E)-1,7-bis(4-hydroxy-3-methoxyphenyl)hepta-1,6-diene-3,5-dione; and Demethoxycurcumin (1E,6E)-1-(4-hydroxy-3-methoxyphenyl)-7-(4-hydroxyphenyl)hepta-1,6-diene-3,5-dione.

**Figure 2 antibiotics-14-00011-f002:**
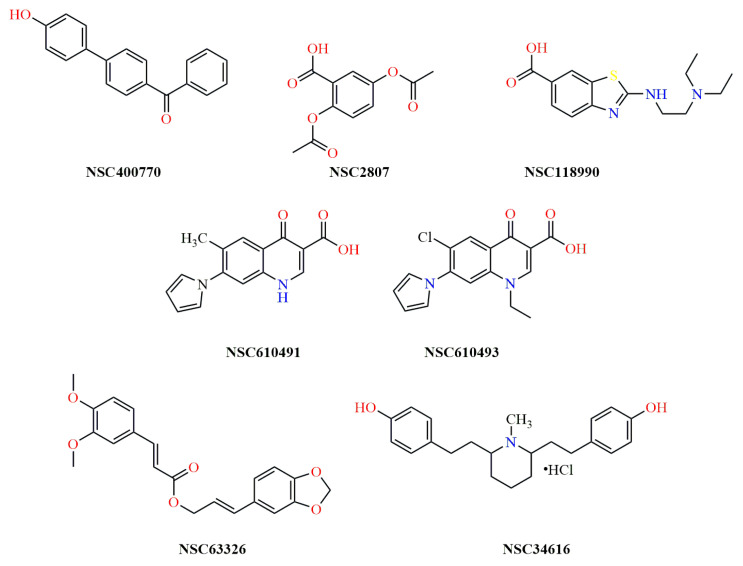
Schematic representation of the NSC compounds from the NCI chemical library.

**Figure 3 antibiotics-14-00011-f003:**
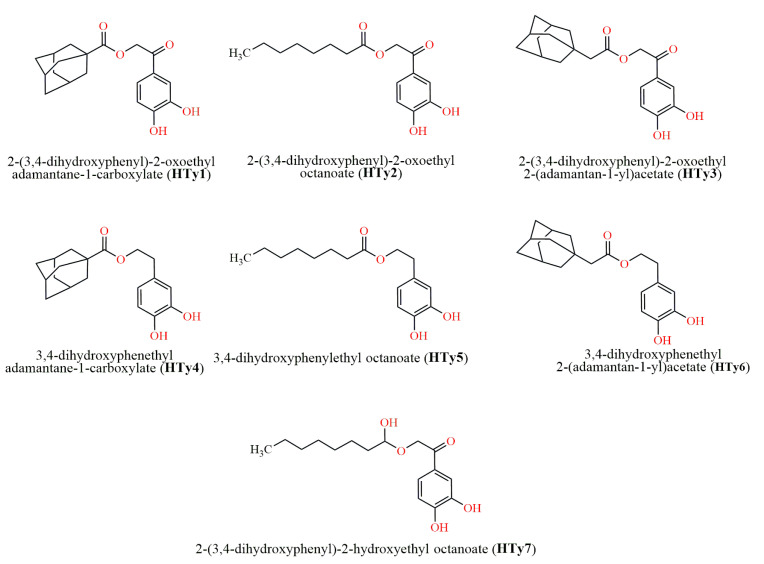
Schematic representation of the synthetic analogues of hydroxytyrosol (HTy) from the Pharmalab chemical library.

**Table 1 antibiotics-14-00011-t001:** NCI hits from in silico similarity screening.

NSC ID	Query Compound	Tanimoto Combo	Antibacterial Activity Reported
2807	CHEMBL423878	1.206	Ν/A
32982	Curcumin	1.725	Antibacterial (CHEMBL116438—Curcumin)
34616	Demethoxycurcumin	1.308	N/A
63326	Demethoxycurcumin	1.208	N/A
118990	CHEMBL151272	1.387	N/A
146617	CHEMBL151272	1.383	Antibacterial (CHEMBL291157)
400770	CHEMBL381395	1.385	Ν/A
610491	CHEMBL151272	1.348	N/A
610492	CHEMBL147475	1.164	Antibacterial (CHEMBL68262)
610493	CHEMBL390983	1.301	N/A
610496	CHEMBL390983	1.329	Antibacterial (CHEMBL68262)
687841	Demethoxycurcumin	1.805	Antibacterial (CHEMBL179512—Demethoxycurcumin)
687844	Curcumin	1.661	Antibacterial (CHEMBL116438—Curcumin)
758614	CHEMBL147475	1.468	Antibacterial (CHEMBL295433)
758701	CHEMBL147475	1.391	Antibacterial (CHEMBL31)
758956	CHEMBL177475	1.433	Antibacterial (CHEMBL6259)
759252	CHEMBL151272	1.406	Antibacterial (CHEMBL430)
759622	CHEMBL390983	1.348	Antibacterial (CHEMBL363449)
759835	CHEMBL151272	1.441	Antibacterial (CHEMBL295619)
759859	CHEMBL147475	1.503	Antibacterial (CHEMBL278255)

NSC, compounds received at the NCI Repository are assigned with an identifying NSC number; N/A, not available.

**Table 2 antibiotics-14-00011-t002:** Pharmalab hits from in silico screening.

Pharmalab ID	Query Compound	Tanimoto Combo	Antibacterial Activity Reported
HTy1	360168	0.972	N/A
HTy2	Demethoxycurcumin	1.052	N/A
HTy3	Demethoxycurcumin	1.005	N/A
HTy4	360168	0.960	N/A
HTy5	Demethoxycurcumin	0.923	N/A
HTy6	360168	0.925	N/A
HTy7	Demethoxycurcumin	0.929	N/A

HTy, hydroxytyrosol; N/A, not available.

**Table 3 antibiotics-14-00011-t003:** Antimicrobial susceptibility testing of selected molecules from the NCI chemical library. The minimum inhibitory concentration (MIC) is expressed in µg/mL.

Compound Name	SA ATCC 25923	MRSA	EF Clinical Isolate VRE	AB	KP Wild Type	PA ATCC 27853	EC ATCC 25922
**NSC 400770**	>100	>100	>100	>100	>100	>100	>100
**NSC 2807**	>100	>100	>100	>100	>100	>100	>100
**NSC 118990**	>100	>100	>100	>100	>100	>100	>100
**NSC 610491**	**25**	**50**	**100**	>100	>100	>100	>100
**NSC 610493**	**12.5**	**12.5**	**100**	>100	**100**	**100**	**100**
**NSC 63326**	**50**	**100**	>100	**100**	>100	>100	>100
**NSC 34616**	>100	>100	>100	>100	>100	>100	>100
**HTy1**	**50**	>100	**50**	>100	>100	>100	>100
**HTy2**	**100**	>100	**25**	>100	>100	**100**	>100
**HTy3**	**50**	>100	**50**	>100	>100	>100	>100
**HTy4**	**50**	>100	**100**	>100	>100	>100	>100
**HTy5**	**50**	>100	**100**	>100	>100	**50**	>100
**HTy6**	**50**	>100	**50**	>100	>100	>100	>100
**HTy7**	>100	>100	**100**	>100	>100	>100	>100

SA, *Staphylococcus aureus*; EF, *Enterococcus faecalis*; VRE, vancomycin-resistant Enterococci; AB, *Acinetobacter baumannii*; KP, *Klebsiella pneumoniae*; PA, *Pseudomonas aeruginosa*; EC, *Escherichia coli*; ATCC, American Type Culture Collection. Best MIC scores are presented in bold.

## Data Availability

Data are available upon request.

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
