# Peer review of "Towards New Scaffolds for Antimicrobial Activity—In Silico/In Vitro Workflow Introducing New Lead Compounds"

_antibiotics, 2024, doi:10.3390/antibiotics14010011_

Round 1

Reviewer 1 Report

Comments and Suggestions for Authors

The authors performed virtual screening of compound libraries and selected 14 compounds for antibacterial activity test. Several compounds were found active against drug-resistant bacteria. Overall, the article presents a promising approach to identifying new antimicrobial compounds using a combination of computational and experimental methods. With further optimization and testing, the identified compounds could potentially contribute to the development of new antimicrobial therapies.

The comments as follows:

The method for preparing the compound entities should be described.

It is better to include  known antibiotics in the activity test as positive control (Figure 3). Is is possible to improve the antibacterial activitis by modifying compound structures? A more explicit discussion on the limitations of the study, such as the scope of the in silico screening or the representativeness of the in vitro testing, would provide a more balanced view.

The resolution of the figure 2 should be improved.

The strain names in the references should be italic.

Author Response

1. Summary

Thank you very much for taking the time to review our manuscript. Please find the detailed responses below and the corresponding revisions/corrections highlighted/in track changes in the re-submitted files.

2. Questions for General Evaluation

Reviewer’s Evaluation

Response and Revisions

Does the introduction provide sufficient background and include all relevant references?

Yes/Can be improved/Must be improved/Not applicable

Thank you for this comment.

Are all the cited references relevant to the research?

Yes/Can be improved/Must be improved/Not applicable

This question is not included in the online form.

Is the research design appropriate?

Yes/Can be improved/Must be improved/Not applicable

Thank you for this comment.

Are the methods adequately described?

Yes/Can be improved/Must be improved/Not applicable

Thank you for the comment. Methodology is now improved taking into consideration your valuable suggestions.

Are the results clearly presented?

Yes/Can be improved/Must be improved/Not applicable

Thank you for the comment. Minor modifications are made to address your suggestion.

Are the conclusions supported by the results?

Yes/Can be improved/Must be improved/Not applicable

This question does not contain an evaluation.

3. Point-by-point response to Comments and Suggestions for Authors

Comments 1: The method for preparing the compound entities should be described.

Response 1: Thank you for pointing this out. We have now added a more specific description of the preparation of our analogs as follows “Growth inhibition was determined in flat-bottom 96-well plates after the prepara-tion of a 2-fold serial dilution of the test analogs. The compounds ordered and shipped from the NCI were in the form of solid powders stored in aliquots. The synthesized compounds were isolated as solid powders and fully characterized by spectroscopic techniques as described elsewhere [31]. All analogs (NSC and HTy) were prepared in 5% dimethyl sulfoxide (DMSO) prior to use and stored at -20°C.” (lines 300-305).

Comments 2: It is better to include known antibiotics in the activity test as positive control (Figure 3). Is is possible to improve the antibacterial activities by modifying compound structures? A more explicit discussion on the limitations of the study, such as the scope of the in silico screening or the representativeness of the in vitro testing, would provide a more balanced view.

Response 2: Thank you for the helpful comment. We have, accordingly, revised some parts of our Discussion section in order to emphasize this point.  

Comments 3: The resolution of the figure 2 should be improved.

Response 3: Thank you for the comment. We did our best to improve the resolution of figure 2.

Comments 4: The strain names in the references should be italic.

Response 4: This comment has been addressed accordingly upon suggestion. We note that we have made the corrections as suggested though we would like to underline that the EndNote is an active online program so there may be difficulties in retaining our corrections made manually.

4. Response to Comments on the Quality of English Language

Point 1:

Response: It is reported by Reviewer 1 that the quality of English does not limit the understanding of the research, therefore only minor corrections are made throughout the text.

5. Additional clarifications

Response: None to be made.

Reviewer 2 Report

Comments and Suggestions for Authors

The compounds selected during the in silico study should be similar to curcumin or demethoxycurcumin, but structural comparison data is lacking.

How are quinolone compounds related to unsaturated keto phenols? e.g. NSC 610491 and NSC 610493?

Why have antibacterial studies not compared curcumin and demethoxycurcumin?

Conclusions are included in the abstract, but not presented separately.

No information was provided on the compounds used in vitro.

Why are NSC ID and Query Compound structural similarities being compared? There is no comparison of biological activities.

I would recommend including an in silico pharmacophore comparison study

Comments on the Quality of English Language

Proofreading required

Author Response

1. Summary

Thank you very much for taking the time to review our manuscript. Please find the detailed responses below and the corresponding revisions/corrections highlighted/in track changes in the re-submitted files.

2. Questions for General Evaluation

Reviewer’s Evaluation

Response and Revisions

Does the introduction provide sufficient background and include all relevant references?

Yes/Can be improved/Must be improved/Not applicable

Thank you for this comment.

Are all the cited references relevant to the research?

Yes/Can be improved/Must be improved/Not applicable

This question is not included in the online form.

Is the research design appropriate?

Yes/Can be improved/Must be improved/Not applicable

Thank you for this comment.

Are the methods adequately described?

Yes/Can be improved/Must be improved/Not applicable

Thank you for the comment. Methodology is now improved taking into consideration your valuable suggestions.

Are the results clearly presented?

Yes/Can be improved/Must be improved/Not applicable

Thank you for the comment. Mοdifications are made to address your suggestion.

Are the conclusions supported by the results?

Yes/Can be improved/Must be improved/Not applicable

Thank you. We have added a small paragraph as conclusions.

3. Point-by-point response to Comments and Suggestions for Authors

Comments 1: The compounds selected during the in silico study should be similar to curcumin or demethoxycurcumin, but structural comparison data is lacking.

Response 1: As it is stated in lines xxx (“In our study, the degree of similarity between the molecules in question (NCI/Pharmalab) and the reference molecules (CHEMBL/CURCUMINS) is evaluated by means of the Tanimoto Combo score which was set as greater than 1.0.”) the selection of the tested analogs was based on the Tanimoto Combo score which is a means to test the degree of similarity between a reference molecule and a molecule in question.

Comments 2: How are quinolone compounds related to unsaturated keto phenols? e.g. NSC 610491 and NSC 610493?

Response 2: Thank you for pointing this out. There is indeed a structural connection between the quinolone skeleton with the unsaturated keto phenols. The latter can be used as starting material towards quinolone synthesis via for example the Cu-catalyzed aza-Michael addition (PMID: 29424545). Consequently, the compounds NSC 610493 and NSC 63326 share the unsaturated ketone moiety in their skeleton. However, the latter lacks the polar and acidic character of the carboxylic substitution of the active quinolone molecule which may be implicated in the enhanced antimicrobial activity of this chemotype. The carboxylic moiety at position 3 is believed to be the primary binding sight of the pharmacophore with the DNA gyrase of the bacterial cell (PMID: 20050522). Hence, lack of this carboxylic moiety in the structure of 63326 results in much reduced potency.

Comments 3: Why have antibacterial studies not compared curcumin and demethoxycurcumin?

Response 3: This question has a good point. In fact, there is an ongoing research referring to antibacterial studies and curcuminoids that remains to be published.

Comments 4: Conclusions are included in the abstract, but not presented separately.

Response 4: Thank you for this comment. We have now added a paragraph as a “Conclusions” part upon suggestion.

Comments 5: No information was provided on the compounds used in vitro.

Response 5: Your comment is now addressed in the Methodology section (lines 300-305).

Comments 6: Why are NSC ID and Query Compound structural similarities being compared? There is no comparison of biological activities.

Response 6: Query compounds were selected based on the MIC activity reported in Chembl without taking into account the biological mechanism of action. Since we intended to expand the structural diversity of antimicrobial compounds by revealing new scaffolds, we selected two databases with compounds not tested on antimicrobial activity.

Comments 7: I would recommend including an in silico pharmacophore comparison study.

Response 7: Thank you for this comment. Since we have a small amount of compounds to create a valid pharmacophore model, we intent to optimize in the future the HTy analogues based on our initial results and thus construct a valid QSAR study. Meanwhile we included in the discussion some main features for Structure Activity Relationships (SAR).

4. Response to Comments on the Quality of English Language

Point 1: Proofreading required.

Response: It is reported by Reviewer 3 that proofreading is required, therefore our text has been revised by an accredited person for professional language editing.

5. Additional clarifications

Response: None to be made.

Round 2

Reviewer 2 Report

Comments and Suggestions for Authors

Thanks for the corrections.